# Wetland Restoration Planning Approach Based on Interval Fuzzy Linear Programming under Uncertainty

**DOI:** 10.3390/ijerph18189549

**Published:** 2021-09-10

**Authors:** Yang Zhang, Jing Shen

**Affiliations:** College of Economics & Management, China Jiliang University, Hangzhou 310018, China; zhy@cjlu.edu.cn

**Keywords:** wetland restoration planning, wetland management, interval fuzzy linear programming, decision-making framework, uncertainty methods

## Abstract

When planning wetland restoration projects, the planting area allocation and the costs of the restoration measures are two major issues faced by decision makers. In this study, a framework based on the interval fuzzy linear programming (IFLP) method is introduced for the first time to plan wetland restoration projects. The proposed framework can not only effectively deal with interval and fuzzy uncertainties that exist in the planning process of wetland restorations but also handle trade-offs between ecological environment benefits and economic cost. This framework was applied to a real-world wetland restoration planning problem in the northeast of China to verify its validity and examine the credibility of the constraints. The optimized results obtained from the framework that we have developed indicate that higher ecological and social benefits can be obtained with optimal restoration costs after using the wetland restoration decision-making framework. The optimal restoration measure allocation schemes obtained by IFLP under different credibility levels can help decision makers generate a range of alternatives, which can also provide decision suggestions to local managers to generate a satisfactory decision-making plan. Furthermore, a comparison was made between the IFLP model and ILP model in this study. The comparison results indicate that the IFLP model provides more information regarding ecological environment and economic trade-offs between the system objective, certainty, and reliability. This framework provides managers with an effective way to plan wetland restoration projects, while transference of the model may help solve similar problems.

## 1. Introduction

Wetlands—transitional zones between aquatic and terrestrial conditions—represent the world’s most productive ecosystems, as not only do they play a critical role in climate regulation, human health, water resource, and biodiversity, but they also provide a series of important ecological services, such as water purification, flood control, and carbon sequestration. At the same time, wetlands are internationally recognized as one of the most vulnerable ecosystems [1,2,3,4,5,6,7]. Approximately 30% of natural wetlands have been lost in recent decades, and the remaining wetlands have suffered severely from climate change and human activities, leading to a dramatic reduction in their extent and widespread degradation [8,9]. Therefore, protecting natural wetlands with the application of restoration planning (which helps wetlands return to a more natural state) is urgently needed to satisfy the needs of human survival and development.

The importance of wetland restoration projects has been emphasized by many studies. Until now, the majority of studies on wetland restoration have focused on restoration measures monitoring [10,11,12], restoration effectiveness evaluation [13,14,15,16,17,18], sustainability prediction [19,20,21], and site selection [22,23]. More specifically, Li et al. (2021) investigated the effects of restoration scenarios on estuarine wetland systems and monitored the wetland restoration measures with a hydrodynamic model [11]. Qu et al. (2018) developed a GIS method and evaluated the wetland restorability of the Sanjiang Plain, China [18]. Horvath et al. (2017) proposed a potential wetland area indicator and used nationally available datasets to identify and prioritize the suitability of wetland restoration for all locations [22]. However, uncertainties may exist in wetland restoration projects, such as the many related costs, impact factors, and objectives, leading to complexities in the relevant decision–analysis processes and difficulties in obtaining optimal strategies with high feasibility and robustness [24]. In wetland ecological management, restoration projects are facing tremendous challenges because of these uncertainties. Therefore, it is necessary to apply practicable and effective optimization methods in wetland restoration projects to address these complex and uncertain conditions. However, very few studies have systematically considered uncertainties in wetland restoration projects and designed an optimum management plan under uncertainty.

The major contribution of this study is the framework for the development of wetland restoration projects, using decision making based on uncertainty methods to plan wetland restoration projects, including a consideration of the planting area allocation and the cost of restoration measures. This is the first framework to introduce the IFLP method into wetland restoration projects and construct an optimized model under uncertainty. This study does not consider land use conversion and related pollution issues in the process of wetland restoration. We applied this framework to a real-world wetland restoration project planning problem in the northeast of China to verify its validity. The results indicated that this framework has advantages in: (1) obtaining a more credible restoration effect with minimized cost; (2) making trade-offs between ecological environment benefits and economic cost under different satisfaction degrees; (3) taking both ecological benefit and social benefit into account; and (4) addressing uncertainties, which featured as interval numbers and fuzzy-interval numbers in the wetland restoration project. With the help of this framework, an optimal allocation pattern of restoration measures can be obtained, which can minimize the restoration costs in wetland ecological management and offer additional ecological and social benefits. The optimal results can help wetland managers formulate more efficient wetland restoration planning strategies.

## 2. Methods

As an extension to interval linear programming (ILP) and fuzzy linear programming (FLP), IFLP can process interval and fuzzy uncertainties that exist in the planning process of wetland restoration. The optimal restoration measure allocation schemes obtained by IFLP under different credibility levels can help decision makers generate a range of alternatives, which can also provide decision suggestions to local managers to generate a satisfactory decision-making plan. Firstly, the formulation and solution algorithm of IFLP was given [25,26,27,28,29,30,31], and then the application of IFLP was introduced in a wetland restoration project. The construction of the wetland restoration decision-making framework based on IFLP is shown in Figure 1.

### 2.1. Interval Fuzzy Linear Programming

An interval linear programming (ILP) model can be defined as follows:(1a)$$Min\;f^{\pm }=\;\;C^{\pm }X^{\pm }$$
subject to:(1b)$$A^{\pm }X^{\pm }\leq B^{\pm }$$
(1c)$$X^{\pm }\geq 0$$
where $A^{\pm }\in {\left\{R^{\pm }\right\}}^{m\times n},\;B^{\pm }\in {\left\{R^{\pm }\right\}}^{m\times 1},\;C^{\pm }\in {\left\{R^{\pm }\right\}}^{1\times n},\;X^{\pm }\in {\left\{R^{\pm }\right\}}^{n\times 1}$, and $R^{\pm }$ denote a set of interval numbers; the ‘−’ and ‘+’ superscripts denote the lower and upper bounds of the parameters/variables, respectively. In model (1), the decision variables ($X^{\pm }$) can be sorted into two categories: continuous and binary.

When the system’s goal and constraints are fuzzy, model (1) can be converted into an interval fuzzy linear programming (IFLP) problem as follows [32]:(2a)$$Max\;f^{\pm }\underset{˜}{=}C^{\pm }X^{\pm }$$
subject to:(2b)$$A^{\pm }X^{\pm }\underset{˜}{\leq }B^{\pm }$$
(2c)$$X^{\pm }\geq 0$$
where symbols $\underset{˜}{=}$ and $\underset{˜}{\leq }$ represent fuzzy equality and inequality, respectively. Based on the principle of fuzzy flexible programming, let *λ^±^* correspond to the membership grade of satisfaction for a fuzzy decision. Specifically, the flexibility in the constraints and fuzziness in the system objective are represented by fuzzy sets and are denoted as ‘fuzzy constraints’ and a ‘fuzzy goal’, respectively. They can be expressed as membership grades [*λ*^±^] corresponding to the degrees of overall satisfaction for the constraints or the objective. Thus, an IFLP model can be formulated as follows:(3a)$$Max\;\lambda^{\pm }$$
subject to:(3b)$$C^{\pm }X^{\pm }\geq f_{}^{+}-\lambda_{}^{\pm }\left(f_{}^{+}-f_{}^{-}\right)$$
(3c)$$A^{\pm }X^{\pm }\leq B^{+}-\lambda_{}^{\pm }\left(B_{}^{+}-B_{}^{-}\right)$$
(3d)$$X^{\pm }\geq 0$$
(3e)$$0\leq \lambda^{\pm }\leq 1$$
where *f*^+^ and *f*^−^ are the upper and lower bounds of the objective’s aspiration level established by decision makers for the objective they desire to achieve, and *λ*^±^ is the control variable corresponding to the membership degree of satisfaction for the fuzzy decision or the constraints.

### 2.2. IFLP Solution Method

The IFLP formulation and its solution algorithm were developed and first used to plan regional solid waste management systems under uncertainty [33]. This model analyzes the detailed interrelationships between the parameters and variables and between the objective function and the constraints. The modeling results can generate a number of decision alternatives under various system conditions, allowing for more in-depth analyses of trade-offs between environmental and economic objectives as well as those between system optimality and reliability. Based on the IFLP solution method, model (3) can be transformed into two deterministic sub-models that correspond to the lower and upper bounds of the objective function value. Generally, the solution procedures for IFLP can be generalized as follows:

Step 1: Transfer model (3) to a sub-model (4), solve it, and obtain solutions for $x_{j\;opt}^{-}$, $x_{j\;opt}^{+}$, $\lambda_{opt}^{+}$ and $f_{opt}^{-}$, respectively. The sub-model (4) is formulated as follows (assuming that $f_{}^{\pm }\geq 0$ and $b_{i}^{\pm }\geq 0$):(4a)$$Max\;\lambda^{+}$$
subject to:(4b)$$\;\sum_{j=1}^{k_{1}}c_{j}^{-}x_{j}^{-}+\sum_{j=k_{1}+1}^{n}c_{j}^{-}x_{j}^{+}\leq f_{}^{+}-\lambda_{}^{+}\left(f_{}^{+}-f_{}^{-}\right)$$
(4c)$$\;\sum_{j=1}^{k_{1}}{\left|a_{ij}\right|}^{+}sign\left(a_{ij}^{+}\right)x_{j}^{-}+\sum_{j=k_{1}+1}^{n}{\left|a_{ij}\right|}^{-}sign\left(a_{ij}^{-}\right)x_{j}^{+}\leq b_{i}^{+}-\lambda_{}^{+}\left(b_{i}^{+}-b_{i}^{-}\right),\forall i$$
(4d)$$x_{j}^{-}\geq 0,\;\;j=1,2,\ldots ..k_{1}$$
(4e)$$x_{j}^{+}\geq 0,\;j=k_{1}+1,\;k_{1}+2,\;\ldots ..,\;n$$
where $x_{j}^{\pm },\;\;j=1,2,\ldots ..k_{1}$ are interval variables with positive coefficients in the objective function, and $x_{j}^{\pm },\;j=k_{1}+1,\;k_{1}+2,\;\ldots ..,\;n$ are interval variables with negative coefficients.

Step 2: Solve sub-model (5) and obtain solutions of $x_{j\;opt}^{+}$, $x_{j\;opt}^{-}$, $\lambda_{opt}^{-}$ and $f_{opt}^{+}$, respectively. According to the solutions of $x_{j\;opt}^{-}$ ($j=1,2,\ldots ..k_{1}$), $x_{j\;opt}^{+}$ ($j=k_{1}+1,k_{1}+2,\ldots ..n$), $\lambda_{opt}^{+}$ and $f_{opt}^{-}$ obtained from sub-model (4), the sub-model corresponding to *f*
^−^ can be formulated as follows (assuming that $f_{}^{\pm }\geq 0$ and $b_{i}^{\pm }\geq 0$):(5a)$$Max\;\lambda^{-}$$
subject to:(5b)$$\;\sum_{j=1}^{k_{1}}c_{j}^{+}x_{j}^{+}+\sum_{j=k_{1}+1}^{n}c_{j}^{+}x_{j}^{-}\leq f_{}^{+}-\lambda_{}^{-}\left(f_{}^{+}-f_{}^{-}\right)$$
(5c)$$\;\sum_{j=1}^{k_{1}}{\left|a_{ij}\right|}^{-}sign\left(a_{ij}^{-}\right)x_{j}^{+}+\sum_{j=k_{1}+1}^{n}{\left|a_{ij}\right|}^{+}sign\left(a_{ij}^{+}\right)x_{j}^{-}\leq b_{i}^{+}-\lambda_{}^{-}\left(b_{i}^{+}-b_{i}^{-}\right),\forall i$$
(5d)$$x_{j}^{+}\geq x_{j\;opt}^{-},j=1,2,\ldots ..,k_{1}$$
(5e)$$0\leq x_{j}^{-}\leq x_{j\;opt}^{+},\;j=k_{1}+1,\;k_{1}+2,\;\ldots ..,\;n$$

The solutions of $x_{j\;opt}^{+}$ ($j=1,2,\ldots ..k_{1}$), $x_{j\;opt}^{-}$($j=k_{1}+1,k_{1}+2,\ldots ..n$), $\lambda_{opt}^{-}$ and $f_{opt}^{+}$ can be obtained using sub-model (5). Thus, the general solutions can be obtained as follows:(6a)$$f_{opt}^{\pm }=[f_{opt}^{-},f_{opt}^{+}]$$
(6b)$$x_{j\;opt}^{\pm }=[x_{j\;opt}^{-},\;\;x_{j\;opt}^{+}]$$
(6c)$$\lambda_{opt}^{\pm }=[\lambda_{opt}^{-},\;\;\lambda_{opt}^{+}]$$

### 2.3. IFLP Model for Wetland Restoration Project

In a wetland restoration project system, results produced by the optimization model can be rendered highly questionable if the modeling inputs are expressed with uncertainty. To reduce the impact of the system uncertainties that are expressed as discrete intervals and fuzzy membership functions on the optimization results, the parameters in the wetland ecological management system can be represented as closed intervals with upper bound and lower bounds and/or fuzzy sets. In this study, the optimization objective of the system is to minimize the wetland restoration costs subject to the planting area allocation and the constraints. The total chloride absorption and ecological benefits during the project implementation period are described by fuzzy numbers with a fuzzy-interval membership function according to their data characteristics. The related parameters are described in Table 1.

The IFLP optimization model can be formulated as follows:(7a)$$Max\lambda^{\pm }$$
subject to:

(1) Project investment constraints
(7b)$$\sum_{i=1}^{I}Q_{i}^{\pm }\cdot L_{i}^{\pm }\cdot x_{i}^{\pm }\geq f{’}^{+}{}_{opt}-(1-\lambda^{\pm })(f{’}^{+}{}_{opt}-f{’}^{-}{}_{opt})$$

(2) Salinization constraints
(7c)$$\sum_{i=1}^{1}h_{i}^{\pm }\cdot c_{i}^{\pm }\cdot M\cdot \omega_{i}^{\pm }\cdot x_{i}^{\pm }\geq N_{0}^{+}-\left(1-\lambda^{\pm }\right)\left(N_{0}^{+}-N_{0}^{-}\right)$$

(3) Carbon sink constraints
(7d)$$\sum_{i=1}^{I}C_{i}^{\pm }\cdot x_{i}^{\pm }\geq C_{o}^{\pm }$$

(4) Total area constraints
(7e)$$\sum_{i=1}^{I}x_{i}^{\pm }\leq A_{0}^{\pm }$$

(5) Labor force constraints
(7f)$$\sum_{i=1}^{I}L_{ki}^{\pm }\cdot x_{i}^{\pm }\leq L_{0}^{\pm }$$

(6) Ecological benefit constraints [34,35,36]
(7g)$$V_{A}^{\pm }+V_{W}^{\pm }+V_{C}^{\pm }\geq V_{0}^{+}-\left(1-\lambda^{\pm }\right)\left(V_{0}^{+}-V_{0}^{-}\right),$$
(7h)$$V_{A}^{\pm }=\sum_{i=1}^{I}P_{Ai}^{\pm }\cdot \beta_{i}^{\pm }\cdot x_{i}^{\pm }$$
(7i)$$V_{W}^{\pm }=\sum_{i=1}^{I}P_{Wi}^{\pm }\cdot \delta_{i}^{\pm }\cdot x_{i}^{\pm }$$
(7j)$$V_{C}^{\pm }=P_{Ci}^{\pm }\cdot \varepsilon_{i}^{\pm }\cdot x_{i}^{\pm }$$

(7) Social benefit constraints
(7k)$$\sum_{i=1}^{I}\alpha_{i}^{\pm }\cdot U_{i}^{\pm }\cdot x_{i}^{\pm }\geq U_{0}^{\pm }$$

(8) Water availability constraints
(7l)$$\sum_{i=1}^{I}W_{i}^{\pm }x_{i}^{\pm }\leq W_{0}^{\pm }$$

(9) Non-negativity constraints
(7m)$$x_{i}^{\pm }\geq 0$$
(7n)$$0\leq \lambda_{}^{\pm }\leq 1$$

## 3. Case Study

### 3.1. Study Area

The shallow basket lake wetland is located in the southwest part of the Songhua River basin (44°22′–44°32′ N, 124°40′–125°59′ E), with a semi-humid continental monsoon climate in the north temperate zone. It lies in the middle of Jilin Province, China, and the western shore forms its border with Siping City. It is the only natural wetland in Jilin Province. With an area of about 300 square kilometers and an average depth of 2 m, it is the third largest inland alkaline freshwater lake in Jilin Province. The shallow basket lake wetland was promoted to a national nature reserve in China in 2011. The length of the surface water is 25 km from north to south and 10 km from east to west. There are various types of wetlands in the shallow basket lake wetland, including lake swamps, reed swamps, and cattail marsh. Therefore, the wetland provides habitat for rare and endangered birds, such as *Grus*
*japonensis*, *Oriental storks, great bustards*, and *sparrowhawks*. It also plays an important role in protecting biodiversity, maintaining a humid climate, maintaining soil and water, and preventing dust and sand, and provides an indispensable ecosystem service function for local residents [37]. The geographic location of the study area can be seen in Figure 2.

The shallow basket lake wetland has a flow cutoff in some rivers because of long-term drought and a shortage of rain. The plummeting groundwater table has also led to a decrease in wetland and grass area, and an increase in groundwater recession, soil erosion, soil desertification, and salinization, which had seriously threatened ecological security. The proportion of the salinization area in the shallow basket lake wetland can be seen in Table 2.

### 3.2. Data Resource

Plants are the basis of wetland ecosystems, as well as wetland restoration. Therefore, plant restoration will be selected as the restoration measure for this wetland restoration project. The vegetation types will be determined considering local resources. The original planning of the wetland restoration project, including tree species, planting mode and planting area, is shown in Table 3. To plan the wetland restoration project in the shallow basket lake, related data were collected from field research, statistical websites, literature surveys, and statistical yearbooks [38,39].

## 4. Results Analysis and Discussion

### 4.1. Optimal Solution of IFLP Model

By inputting the basic data into the developed IFLP model of the wetland restoration decision-making framework, the allocation pattern of wetland restoration measures can be obtained. Table 4 presents the optimal solutions for the allocation pattern of restoration measures obtained from the IFLP model, including the allocation of a mixed forest planting pattern and a pure forest planting pattern. Note that the optimal solutions for the objective function value and most of the non-zero decision variables related to the wetland restoration measures are interval, while wetland restoration measures related to *Populus bolleana* are deterministic values. The solution indicated that the optimal planting area of reeds was (47.02, 48.36) km^2^ during the project implementation period. It also suggested that the mixed forest had two patterns with P&D, as well as *Populus euphratica* & *Elaeagnus angustifolia* (P&E), with a planting area of (32.69, 37.07) km^2^ and (36.59, 38.37) km^2^, respectively. Meanwhile, *Populus bolleana* were suggested to be planted with an area of 12.81 km^2^. From the optimal allocation pattern of restoration measures calculated from the IFLP model, the total restoration cost would be (2216.97, 2403.42) (10^4^ CNY), with the degree of overall satisfaction ($\lambda^{\pm }$) being (0.36, 0.91). In general, the optimal solutions obtained from the IFLP model are presented with an upper bound and a lower bound. These interval results from the wetland restoration optimization model indicate that the final decisions are sensitive to uncertain inputs from project managers. In contrast, certain solutions from the wetland restoration optimization model with the traditional method are not sensitive to the input uncertainties. Thus, alternative schemes of the wetland restoration optimization project can be achieved by adjusting the interval solutions in the range of its lower and upper bounds according to the various project management requirements. For example, the solutions of $x_{3}^{\pm }$ under the given constraints reflect intervals of planting area for *Populus euphratica* and *Elaeagnus angustifolia* with mixed forest planting pattern. The upper bound of $x_{b}^{\pm }$ (i.e., $x_{b}^{+}$) corresponds to a higher objective result, and the lower bound of $x_{b}^{\pm }$ (i.e., $x_{b}^{-}$) corresponds to a lower objective result.

The solution of the wetland restoration project from model (7) would be [2216.97, 2403.42] (10^4^ CNY), with the $\lambda^{\pm }$ range being [0.36, 0.91]. The lower system cost represents an alternative with a lower project investment for wetland restoration, and vice versa. In general, planning with a higher cost would guarantee that the main wetland restoration objective and the ecological requirements will be met; conversely, if the plan aims toward a lower cost, there may be risks of violating these requirements. The $\lambda^{\pm }$ level ($\lambda^{\pm }$ = [0.36, 0.91]) represents the possibility of satisfying the main objective and the constraints. It corresponds to the decision makers’ preference regarding ecological environment and economic trade-offs. Specifically, $\lambda^{-}$ corresponds to a higher system cost ($f_{opt}^{+}$) under demanding conditions; $\lambda^{+}$ is related to $f_{opt}^{-}$ under advantageous conditions. The lower bound of $\lambda^{\pm }$ ($\lambda^{-}$) is merely 0.36, which indicates a relatively low possibility of satisfying the objective function and the constraints. In addition, from an interval solution perspective, it can be seen that the solution provided the interval range of total wetland restoration project investment under the optimal allocation pattern of restoration measures. As the actual value of each system variable or input parameter could be any value in its interval, the total investment of the wetland restoration project would fluctuate between $f_{opt}^{-}$ and $f_{opt}^{+}$ as the system variables changed. An allocation pattern of wetland restoration measures with a lower bound represents a lower wetland restoration cost, and vice versa. Therefore, the optimal allocation pattern of wetland restoration measures generates a number of decision alternatives under different system conditions.

### 4.2. Discussion

More details of the allocation pattern for wetland restoration measures between original scheme and the IFLP scheme were analyzed. Figure 3 presents the wetland restoration measures under a different scheme. The total restoration cost in original project plan was 2593.41 (10^4^ CNY). In comparison, it was (2216.97, 2403.42) in the optimal solution, which can save (189.99, 376.44) (10^4^ CNY) in restoration costs. Thus, the range of 7.33–14.52% cost reduction verified the effectiveness and validity of this optimization model.

The benefits between the original project plan and the optimization model are different (shown in Figure 4). Compared with the original scheme, the optimal solution based on the wetland restoration decision-making framework can lead to better ecological and social benefits with less investment. The results obtained from the IFLP model implied that the ecological benefits and social benefits had a positive correlation with restoration costs. The possibility of satisfying the system objective and the constraints level was $\lambda^{\pm }$ = (0.36, 0.91). Ecological benefits (soil containment and water conservation, water purification, and microclimate regulation) and social benefits in the optimization results from the IFLP method were (3099.98, 3342.29), (2752.03, 3339.02), (8775.68, 9134.37) and (9942.11, 11042.54) (10^4^ CNY), respectively. In comparison to the original scheme, the interval optimal solutions had corresponding (10.53%, 19.17%), (6.26%, 28.92%), (9.15%, 13.61%), and (16.67%, 29.58%) increases. In conclusion, lower restoration costs can bring higher ecological environment and social benefits after applying a practicable and optimized planting allocation pattern based on the wetland restoration decision-making framework. This conclusion is also in accordance with the results from previous research [37,40].

The solutions for an allocation pattern of wetland restoration measures can also be solved through an ILP model that expresses uncertainties as intervals [40]. However, as the system variables and input parameters are interval values, the main limitation of the ILP model is its over-simplification of fuzzy membership information into intervals. Hence, the obtained allocation pattern of restoration measures based on an ILP model lacks system reliability information as defined by $\lambda_{opt}^{\pm }$. Figure 5 presents an objective function value comparison of the wetland restoration project obtained through the ILP and IFLP approaches. The optimal solutions obtained from the ILP model provide a total project investment of (2193.14, 2416.01) (10^4^ CNY), whereas the IFLP model leads to a total project investment of (2216.97, 2403.42) (10^4^ CNY), with the possibility of satisfying the system objective and the constraints level being $\lambda^{\pm }$ = (0.36, 0.91). It can be seen that the IFLP model results in a higher mid value and a smaller interval than the ILP model. The raised benefit corresponds to a reduced possibility of satisfying the objective and the constraints; the increased system certainty (i.e., the shrunk interval width) is based on a reduced certainty of the possibility of satisfying the objective and the constraints. Thus, the IFLP approach provides more information regarding ecological environment and economic trade-offs between the system objective, certainty, and reliability [29,41]. When the interval range for interval system parameters and fuzzy function for fuzzy uncertainty system parameters changed, the wetland restoration cost simultaneously fluctuated between $f_{opt}^{-}$ and $f_{opt}^{+}$ with a variety of reliability levels.

Compared with the traditional optimization methods and ILP methods [40], the IFLP approach applied in a wetland restoration decision-making framework has superiority in a number of ways. Firstly, the quality of available wetland restoration information for system modelling is often not good enough in most cases to be presented as deterministic numbers. Instead, some uncertainty data can only be quantified as intervals or vague values. The IFLP can effectively handle various uncertainties described as possibility distributions and intervals that exist in the system variables or input parameters. Secondly, the model for wetland restoration projects based on an IFLP approach can directly incorporate uncertainties within its optimization framework. Its optimal solutions are presented with combinations of deterministic, interval, and distribution information, offering flexibility in the interpretation of results and decision–alternative generation. Thus, the project managers can obtain a satisfactory wetland management plan according to the practical situation and level of risk.

## 5. Conclusions

In this study, a wetland restoration decision-making framework based on an IFLP method is proposed for the first time to optimize planting area allocation and restoration costs. This developed framework incorporates interval programming, fuzzy programming, and fuzzy-interval numbers in a general framework to optimize limited investment. Trade-offs between ecological environment benefits and economic benefits under different reliability levels were fully considered through this framework. Furthermore, the framework can handle uncertainties featured as interval numbers and fuzzy-interval numbers.

This approach is applied to a real case study in the shallow basket lake wetland, Songhua River basin in China, to verify its validity and examine the credibility of the constraints. The optimized results obtained from the framework show that lower restoration costs can bring higher ecological environment and social benefits after applying a practicable and optimized planting allocation pattern based on the wetland restoration decision-making framework. The optimal restoration measure allocation schemes obtained from the IFLP under different credibility levels can help decision makers generate a range of alternatives, which can also provide decision suggestions to local managers to generate a satisfactory decision-making plan. The optimal restoration measure allocation schemes obtained by the IFLP under different credibility levels can help decision makers generate a range of alternatives.

The framework and models developed in this study are portable. According to this framework, the relationship between economic costs and ecological environment benefits can be analyzed. This framework and thinking can be applied to other wetland areas to address planting area allocation and restoration cost optimization problems. In future research, this method can take a range of detailed information into account, such as the impacts of climate change and the market price changes of plants, to construct a robust and comprehensive wetland restoration project decision-making framework. 

## Figures and Tables

**Figure 1 ijerph-18-09549-f001:**
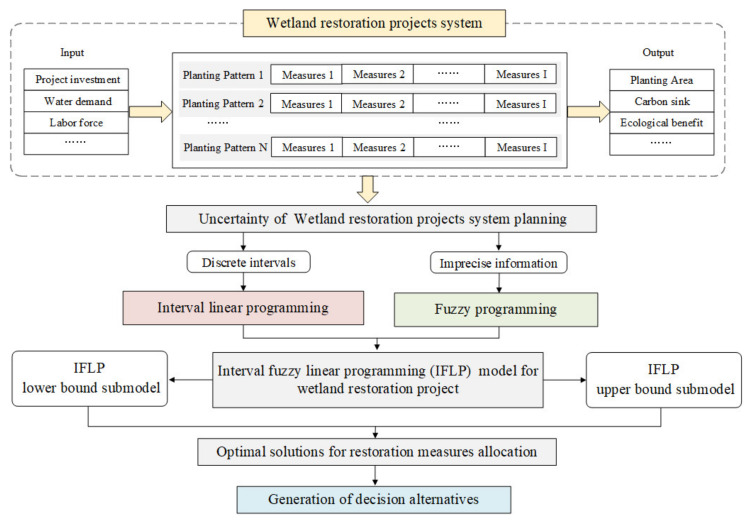
Framework of the developed IFLP model for a wetland restoration project.

**Figure 2 ijerph-18-09549-f002:**
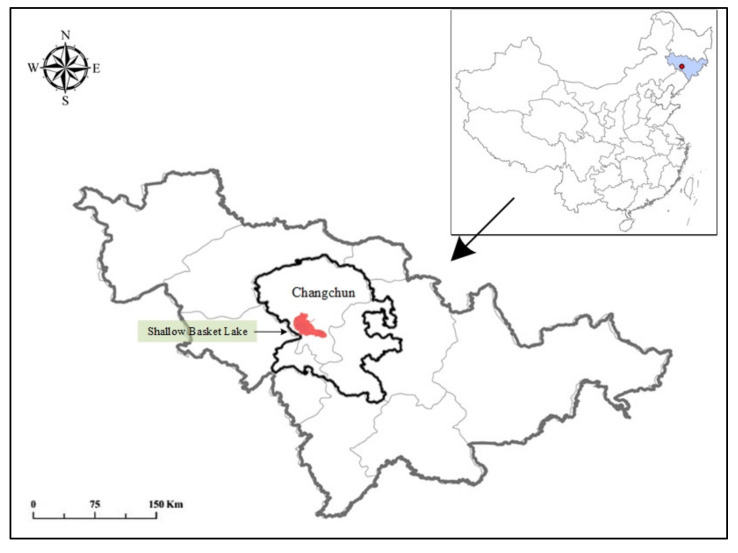
The geographic location of the study area.

**Figure 3 ijerph-18-09549-f003:**
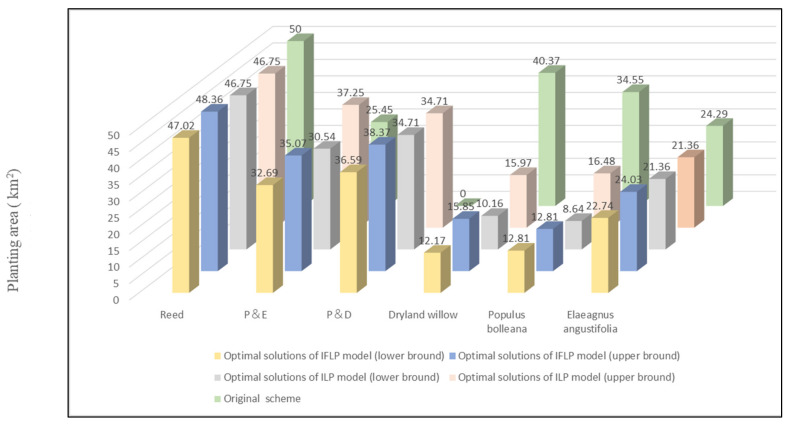
A comparison of planting areas between the original project plan and the optimization model.

**Figure 4 ijerph-18-09549-f004:**
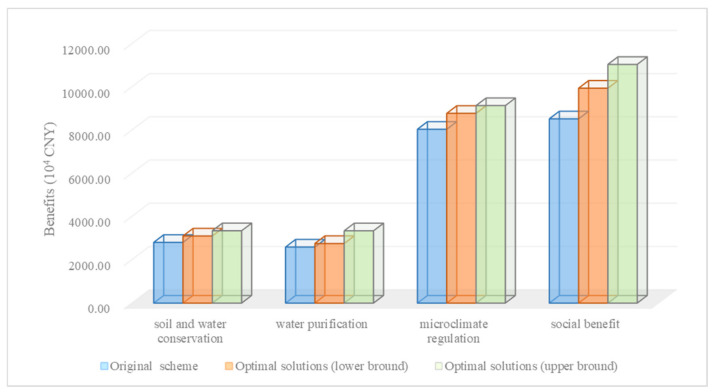
A comparison of benefits between the original project plan and the optimization model.

**Figure 5 ijerph-18-09549-f005:**
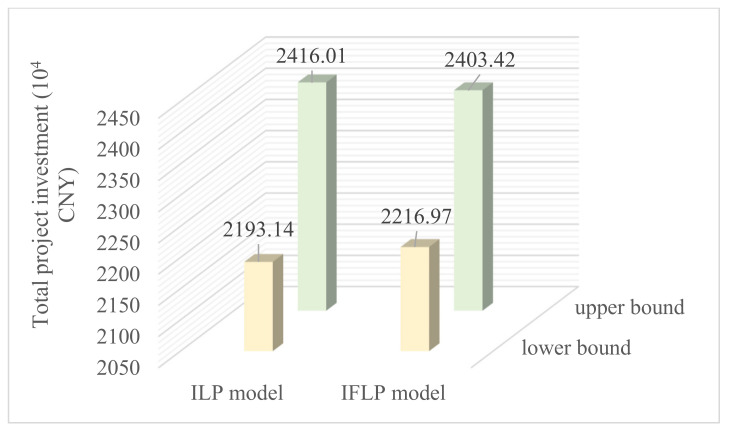
Comparison of total project investment through ILP and IFLP models.

**Table 1 ijerph-18-09549-t001:** Definitions of symbols in the IFLP model.

Symbol	Definition
*f^±^*	Total expected system cost (10^4^ CNY)
*i*	Type of restoration measures
$\lambda^{\pm }$	Control variable corresponding to the degree of satisfaction for the fuzzy objective or the constraints
$Q_{i}^{\pm }$	Cost of saplings in restoration measures *i* (10^4^ CNY/plant)
$L_{i}^{\pm }$	Planting density in the *i*th restoration measures (plant/km^2^)
$x_{i}^{\pm }$	Areas of *i*th restoration measures (km^2^)
$m_{i}^{\pm }$	Ecological water demand quota of restoration measures *i* (10^4^ m^3^/km^2^)
$W_{l}^{\pm }$	Water requirement of the lake (10^4^ m^3^)
$W_{m}^{\pm }$	Water requirement of the marsh (10^4^ m^3^)
$W_{s}^{\pm }$	Water requirement of the soil (10^4^ m^3^)
$W_{a}^{\pm }$	Water requirement of the wildlife habitat (10^4^ m^3^)
$W_{o}^{\pm }$	Maximum total water requirement in the wetland (10^4^ m^3^)
$h_{i}^{\pm }$	Thickness of planting soil layer in restoration measures *i* (m)
$c_{i}^{\pm }$	Chloride concentration in the *i*th restoration measures (mol/m^3^)
*M*	Molecular weight of chloride (g/mol)
$\omega_{i}^{\pm }$	Absorption coefficient of chloride ion in restoration measures *i*
$N_{0}^{\pm }$	Chloride ion absorption in the planning period (tonnes)
$C_{i}^{\pm }$	Capacity per unit area of vegetation type in restoration measures *i* (tonnes/km^2^)
$C_{o}^{\pm }$	Total amount of carbon sink (tonnes)
$A_{o}^{\pm }$	Area of the wetland restoration project (km^2^)
$L_{ki}^{\pm }$	Available labor coefficient per unit area of planting type in restoration measures *i* (man-day/ha)
$L_{0}^{\pm }$	Total labor force (man-day)
$V_{A}^{\pm }$	Benefits of microclimate regulation
$V_{W}^{\pm }$	Benefits of water purification
$V_{C}^{\pm }$	Benefits of soil and water conservation
$V_{0}^{\pm }$	Total ecological benefits
$P_{Ai}^{\pm }$	Savings in electricity consumption due to temperature regulation by restoration measures *i* (10^4^ CNY/km^2^)
$\beta_{i}^{\pm }$	Correction factor for calculating benefits of microclimate regulation by restoration measures *i*
$P_{Wi}^{\pm }$	Savings in sewage treatment cost by restoration measures *i* (10^4^ CNY/km^2^)
$\delta_{i}^{\pm }$	Correction factor for calculating benefits of water source purification by restoration measures *i*
$P_{Ci}^{\pm }$	Savings in cost of soil and water conservation by restoration measures *i* (10^4^ CNY/km^2^)
$\varepsilon_{i}^{\pm }$	Correction factor for calculating benefits of soil and water conservation by restoration measures *i*
$U_{i}^{\pm }$	Economic benefit of plants by restoration measures *i* (10^4^ CNY/tonnes)
$\alpha_{i}^{\pm }$	Yield per unit area of economic plants by restoration measures *i* (tonnes/km^2^)
$U_{0}^{\pm }$	Total economic benefit in the planning period (10^4^ CNY)

**Table 2 ijerph-18-09549-t002:** Proportion of the salinization area in the shallow basket lake wetland.

Salinization Degree	Salinization Area	Proportion
Slight salinization	1800.59	49.33
Medium salinization	833.82	22.84
High salinization	1015.63	27.83

**Table 3 ijerph-18-09549-t003:** The original scheme of the wetland restoration project.

Planting Mode	Restoration Measures	Planting Area (km^2^)
Mixed forest	*Populus euphratica*, dryland willow (P&D)	25.45
Pure forest	Reed (*Phragmites karka*)	50
	Dryland willow	40.37
	*Populus bolleana*	34.55
	*Elaeagnus angustifolia*	24.29
Total project investment (10^4^ CNY) = 2593.38		

**Table 4 ijerph-18-09549-t004:** Optimal solutions for allocation pattern of restoration measures obtained from the IFLP model.

Planting Mode	Restoration Measures	Planting Area (km^2^)	Symbol
Mixed forest	*Populus euphratica*, dryland willow	(32.69, 37.07)	$x_{a}^{\pm }$
	*Populus euphratica*, *Elaeagnus angustifolia*	(36.59, 38.37)	$x_{b}^{\pm }$
Pure forest	Reed (*Phragmites karka*)	(47.02, 48.36)	$x_{c}^{\pm }$
	Dryland willow	(12.17, 15.85)	$x_{d}^{\pm }$
	*Populus bolleana*	12.81	$x_{e}^{\pm }$
	*Elaeagnus angustifolia*	(22.74, 24.03)	$x_{f}^{\pm }$
$\lambda^{\pm }$ = [0.36, 0.91]			
Total project investment (10^4^ CNY): $f_{opt}^{\pm }$ = (2216.97, 2403.42]			

## Data Availability

The data presented in this study are available on request from the corresponding author.

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
