# Peer review of "Wetland Restoration Planning Approach Based on Interval Fuzzy Linear Programming under Uncertainty"

_ijerph, 2021, doi:10.3390/ijerph18189549_

Round 1
Reviewer 1 Report
This manuscript addresses the wetland restoration projects in a watershed scale of Songhua River basin, Jilin Province, a northeast regions in PR China, to simulate with fuzzy linear programming (IFLP), which is still relatively interesting and is of great current concern to those dealing with real-world data associated with their symbols as well as the watershed quantity/quality from landscape scales. I found the manuscript sufficiently original and interesting to warrant future publication after it is revised. In general, the manuscript is well written. The structure and presentation of the paper is clear. Case study is well-executed and a good demonstration of the viability and usefulness of the model for watershed planning. Article is clearly written and well organized. Literature review is extensive, and references are appropriate. However, the novelty of the work is not present in a good way. The authors need to modify the introduction part and highlight the novelty, as well as the objective of the study. There is one specific question: How the authors get the driving force from the symbols from lowland landscapes in response to the functions of wetland restorations? Do you have QA/QC in your programming process? How did you correct this program (IFLP) to be well- proofed? Why did you not want to study polluted and/or non-polluted lands (if you have) from land conversions in wetland restorations since you used the data of tracing many social-economic parameters associated with water quality at study areas? Any differences of specialty from these unique types of lands in wetland restoration projects in China? You need more explanation for our international readers.
Introduction
Please add your hypothesis through your objective of this study in wetland restoration project. It needs to be more clearly defined and how the results may aid research in a wider context in your model.
Materials and Methods
Your modelling in your detecting regarding to trace many signals in-situ wetlands, and please announce the variation of thresholds as well for some non-polluted lands and polluted lands, such as wetland restorations in wet/open water areas, and urban areas in river basins, should be considered to measure the relations of agricultural fertilizers and pesticides, overgrazed and opened up wasteland, and/or release of factories (may be do not have) from the perspective for each connected riparian corridors and waterbodies in a substantive dynamic model. Any model could be involved in these aforementioned data? Any anthropogenic data and/or natural process could be used? How to measure land-use change in your individual land conversion is crucial to calculate in your landscape metric analyses. If you consider class-level metrics (i.e., metrics that apply to land use classes), then, as I said I am interested on your process simulation: “…The development of agriculture and animal husbandry had exceeded the carrying capacity of wetlands and accelerated the drying up of wetlands(page 8).” How to measure the relations between agricultural lands and your tracing in pollutants and/or water scarcity? Do you have global warming issues regarding to drying up in wetlands? How about natural process? And why? Any conflicts with your arguments? Any limitation in your model? Please be specific.
Case Study
1) Why “the shallow basket lake wetland has serious salinization due to long-term drought and shortage of rain that had severely threatened wetland ecological environment? (page 8)” Any particular reasons for this conversion from agricultural lands and/or rangelands more than that of non-agricultural lands in the entire study areas to contribute the loading of your tracing upon this serious salinization? I need to figure out the relations between the pollutants/droughts of agricultural lands and rangelands.
2) More information should be provided about your management approach related to this specific “watershed priority area” since this area is close to the City of Changchun, Jilin Province, a northeast regions in PR China. What policy regarding to your recommendations should be made from this Land Use and Land Cover data (LULC)-based analysis in your fuzzy linear programming (IFLP)?
3) What is your means of “optimized planting allocation pattern” (page 13) from your recommendation? And please be specific.
4) Please be specific to detect your limitations on your model and, all needed future developments could be requested clearly addressed.
Author Response
The authors would like to express their sincere appreciation to the editor and the reviewers for their careful reading of the paper and constructive comments. We have taken these comments into account in our revision. The list of changes and our reply to the reviewers are given as an attachment.

Reviewer 2 Report
The article has a sound introduction, correctly described methodology. The discussion lacks reference to other authors' work and their findings. The advantages and disadvantages of the method are worth demonstrating against other studies. Furthermore, results without the discussion that is typical for scientific articles are not trustworthy. The bibliography contains current scientific articles from the research field. The article requires editorial improvement, there are different font sizes in sentences and other shortcomings.
Figure 1 - "labor force" convert to "Labor force"
Figures 3, 4 and 5 should be quite coherent graphically
Author Response

(The authors gave the same response as above.)

Round 2
Reviewer 1 Report
This revised manuscript that is satisfactory is acceptable to me and fulfils a particular need or purpose in publication for MDPI. Congratulations!